# State Advantage Weighting for Offline RL

**Jiafei Lyu**[1,*] **Aicheng Gong**[1,4]**, Le Wan** [3]**, Zongqing Lu**[2]**, Xiu Li**[1]
[1]Tsinghua Shenzhen International Graduate School, Tsinghua University
[2]School of Computer Science, Peking University
[3]IEG, Tencent
[4]China Nuclear Power Engineering Company Ltd
{lvjf20, gac19}@mails.tsinghua.edu.cn, vinowan@tencent.com,
zongqing.lu@pku.edu.cn, li.xiu@sz.tsinghua.edu.cn

## Abstract

We present *state advantage weighting* for offline reinforcement learning (RL). In contrast to action advantage $A(s, a)$ that we commonly adopt in QSA learning, we leverage state advantage $A(s, s')$ and QSS learning for offline RL, hence decoupling the action from values. We expect the agent can get to the high-reward state and the action is determined by how the agent can get to that corresponding state. Experiments on D4RL datasets show that our proposed method can achieve remarkable performance against the common baselines. Furthermore, our method shows good generalization capability when transferring from offline to online.

## 1 Introduction

Offline reinforcement learning (offline RL) generally defines the task of learning a policy from a static dataset, which is typically collected by some unknown process. This setting has aroused wide attention from the community due to its potential for scaling RL algorithms in real-world problems.

One of the major challenges in offline RL is extrapolation error [16, 26], where the out-of-distribution (OOD) actions are overestimated. Such error is accumulated through bootstrapping, which in turn negatively affects policy improvement. Prior methods address this problem via either making the learned policy stay close to the data-collecting policy (behavior policy) [15, 26, 54], learning without querying OOD samples [24, 57], explicitly assigning (low) values to OOD actions [27, 32], leveraging uncertainty measurement [22, 59, 55, 2], etc.

In this paper, we instead explore a novel QSS-style learning paradigm for offline RL. Specially, we estimate the state $Q$-function $Q(s, s')$, which represents the value of transitioning from the state $s$ to the next state $s'$ and acting optimally thenceforth: $Q(s, s') = r(s, s') + \gamma \max_{s'' \in \mathcal{S}} Q(s', s'')$. By doing so, we decouple actions from the value learning, and the action is determined by how the agent can reach the next state $s'$. The source of OOD will then turn from next action $a'$ into next next state $s''$. In order to get $s''$, we additionally train a predictive model that predicts the feasible and high-value state. We deem that this formulation is more close to the decision-making of humans, e.g., we predict where we can go and then decide how we can get there when climbing.

Unfortunately, we find that directly applying D3G [12], a typical QSS-learning algorithm, is infeasible in offline settings. We wonder: *can QSS learning work for offline RL?* Motivated by IQL [24], we propose to learn the value function by expectile regression [23] such that both the state $Q$-function $Q(s, s')$ and value function $V(s)$ can be well-trained. We train extra dynamics models for predicting the next next state $s''$. We train an *inverse dynamics model* $I(s, s')$ to determine the action, i.e., how to reach $s'$ from $s$. We leverage *state advantage* $A(s, s') = Q(s, s') - V(s)$, which describes how the

---

[*]Work done while working as an intern at Tencent IEG.

next state $s'$ is better than the mean value, for weighting the update of the actor and the model. To this end, we propose **S**tate **A**dvantage **W**eighting (SAW) algorithm. We conduct numerous experiments on the D4RL benchmarks. The experimental results indicate that our method is competitive or even better than the prior methods. Furthermore, we demonstrate that our method shows good performance during online learning, after the policy is initialized offline.

## 2 Preliminaries

We consider an environment that is formulated by a Markov Decision Process (MDP) $\langle \mathcal{S}, \mathcal{A}, \mathcal{R}, p, \gamma \rangle$, where $\mathcal{S}$ denotes the state space, $\mathcal{A}$ represents the action space, $\mathcal{R}$ is the reward function, $p$ is the transition dynamics, and $\gamma$ the discount factor. In QSA learning, the policy $\pi : \mathcal{S} \mapsto \mathcal{A}$ determines the behavior of the agent. The goal of the reinforcement learning (RL) agent is to maximize the expected discounted return: $\mathbb{E}_\pi[\sum_{t=0}^\infty \gamma^t r_{t+1}]$. The action $Q$-function describes the expected discounted return by taking action $a$ in state $s$: $Q^\pi(s,a) = \mathbb{E}_\pi[\sum_{t=0}^\infty \gamma^t r_{t+1}|s_0 = s, a_0 = a]$ The action advantage is defined as: $A(s,a) = Q(s,a) - V(s)$, where $V(s)$ is the value function. The Q-learning gives:

$$Q(s,a) \leftarrow Q(s,a) + \alpha[r + \gamma \max_{a' \in \mathcal{A}} Q(s',a') - Q(s,a)]. \tag{1}$$

The action is then decided by $\arg\max_{a \in \mathcal{A}} Q(s,a)$. In QSS learning, we focus on the state $Q$-function: $Q(s,s')$. That is, the value in QSS is independent of actions. The action is determined by an inverse dynamics model $a = I(s,s')$, i.e., what actions the agent takes such that it can reach $s'$ from $s$, $\pi : \mathcal{S} \times \mathcal{S} \mapsto \mathcal{A}$. We can similarly define that the optimal value satisfies $Q^*(s,s') = r(s,s') + \gamma \max_{s'' \in \mathcal{S}} Q^*(s',s'')$. The Bellman update for QSS gives [12]:

$$Q(s,s') \leftarrow Q(s,s') + \alpha[r + \gamma \max_{s'' \in \mathcal{S}} Q(s',s'') - Q(s,s')]. \tag{2}$$

We further define the *state advantage* $A(s,s') = Q(s,s') - V(s)$, which measures how good the next state $s'$ is over the mean value.

## 3 SAW: State Advantage Weighting for Offline RL

In this section, we first experimentally show that directly applying a typical QSS learning algorithm, D3G [12], results in a failure in offline RL. We then present our novel offline RL method, SAW, which leverages the state advantage for weighting the update of the actor and the prediction model.

### 3.1 D3G Fails in Offline RL

As a typical QSS learning algorithm, D3G [12] aims at learning a policy with the assumption of deterministic transition dynamics. In addition to the state $Q$-function, it learns three models, a prediction model $M(s)$ that predicts the next state with the current state as the input; an inverse dynamics model $I(s,s')$ that decides how to act to reach $s'$ starting from $s$; a forward model $F(s,a)$ that receives state and action as input and outputs the next state, making sure that the proposed state by the prediction model can be reached in a single step. The prediction model, inverse dynamics model (actor), and forward model are all trained in a supervised learning manner. Unfortunately, D3G exhibits very poor performance on continuous control tasks with its vanilla formulation (e.g., Walker2d-v2, Humanoid-v2). We then wonder: will D3G succeed in offline settings?

We examine this by conducting experiments on hopper-medium-v2 from D4RL [14] MuJoCo datasets. We observe in Figure 1(a) that D3G fails to learn a meaningful policy on this dataset. As shown in Figure 1(b), the $Q$ value (i.e., $Q(s,s')$) is extremely overestimated (up to the scale of $10^{12}$). We then wonder, which is the key intuition of this paper, *can we make QSS learning work in offline RL?* This is important due to its potential for promoting learning from observation and goal-conditioned RL in the offline manner.

To this end, we propose our novel QSS learning algorithm, **S**tate **A**dvantage **W**eighting (SAW). We observe that our method, SAW, exhibits very good performance on hopper-medium-v2, with its value estimated fairly well, as can be seen in Figure 1(c).

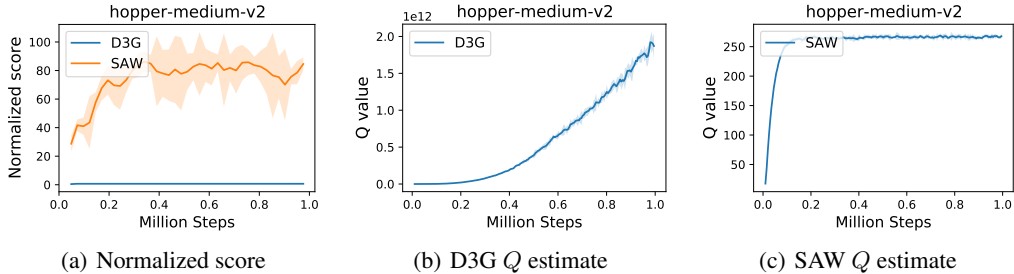

| (a) Normalized score | (b) D3G $Q$ estimate | (c) SAW $Q$ estimate |

Figure 1: Normalized score comparison of D3G against our method on hopper-medium-v2 from D4RL (a). The $Q$ value estimate of D3G incurs severe overestimation (b) while our SAW does not (c). The results are obtained over 5 random runs, and the shaded region captures the standard deviation.

## 3.2 State Advantage Weighting

Under the novel framework of QSS learning, we also aim at learning $Q(s, s')$. To boost the stability of the value estimate and avoid overestimation, we leverage the state advantage $A(s, s')$ instead of the action advantage $A(s, a)$. Our method is motivated by IQL [24], which learns entirely within the support of the dataset. IQL trains the value function $V(s)$ using a neural network, and leverages expectile regression for updating the critic and (action) advantage weighted regression for updating the actor. Similarly, we adopt expectile regression for the critic and (state) advantage weighted regression for updating the prediction model and the actor.

To be specific, we need to train four extra parts other than the critic, a value function $V(s)$, a forward dynamics model $F(s, a)$, a prediction model $M(s)$, and an inverse dynamics model $I(s, s')$ (the actor). The critic we want to learn is updated via expectile regression, which is closely related to the quantile regression [34]. The expectile regression gives:

$$\arg\min_{m_\tau} \mathbb{E}_{x \sim X}[L_2^\tau(x - m_\tau)], \tag{3}$$

where $L_2^\tau(u) = |\tau - \mathbb{I}(u < 0)|u^2$, $\mathbb{I}$ is the indicator function, $X$ is a collection of some random variable. This loss generally emphasizes the contributions of $x$ values larger than $m_\tau$ and downweights those small ones. To ease the stochasticity from the environment (identical to IQL), we introduce the value function and approximate the expectile with respect to the distribution of next state, i.e.,

$$\mathcal{L}_\psi = \mathbb{E}_{s,s' \sim \mathcal{D}}[L_2^\tau(Q_{\theta'}(s, s') - V_\psi(s))], \tag{4}$$

where the state $Q$-function is parameterized by $\theta$ with a target network parameter $\theta'$, and the value function is parameterized by $\psi$. Then, the state $Q$-functions are updated with the MSE loss:

$$\mathcal{L}_\theta = \mathbb{E}_{s,s' \sim \mathcal{D}}[(r(s, s') + \gamma V_\psi(s') - Q_\theta(s, s'))^2]. \tag{5}$$

Note that in Equation (4) and (5), we only use state and next state from the fixed dataset to update the state $Q$-function and value function, leaving out any worry of bootstrapping error.

**Training the forward model.** The forward model $F_\phi(s, a)$ parameterized by $\phi$ receives the state and action as input and predicts the next state (no reward signal is predicted). A forward model is required as we want to ensure that the proposed state by our method is reachable in one step. To be specific, if we merely train one forward model $f(s)$ that predicts the next state based on the current state, there is every possibility that the proposed state is unreachable, inaccurate, or even invalid. However, if we train a forward model to predict the possible next state and encode that information in the prediction model, it can enhance the reliability of the predicted state. The forward model is trained by minimizing:

$$\mathcal{L}_\phi = \mathbb{E}_{s,a,s' \sim \mathcal{D}}\|F_\phi(s, a) - s'\|_2^2. \tag{6}$$

**Training the reverse dynamics model.** We also need the reverse dynamics model $I_\omega(s, s')$ parameterized by $\omega$ to help us identify how the agent can reach the next state $s'$ starting from the current state $s$. The inverse dynamics model is trained by weighted imitation learning, which is similar in spirit to advantage weighted regression (AWR) [40, 24, 50, 39, 37]:

$$\mathcal{L}_\omega = \mathbb{E}_{s,a,s' \sim \mathcal{D}}\left[\exp\left(\beta A(s, s')\right)\|I_\omega(s, s') - a\|_2^2\right], \tag{7}$$

where $\beta \in [0, +\infty)$ is the temperature, and $A(s, s') = Q(s, s') - V(s)$ is the state advantage. By doing so, we downweight those bad actions and prefer actions that incur high reward states.

**Training the prediction model.** In the formulation of QSS, we need to evaluate the next next state $s''$. Therefore, an additional prediction network $M(s)$ is required, i.e., $s'' = M(s')$. It is critical to output a good $s''$ in QSS learning. We want that the agent can reach a high reward $s''$. To fulfill that, we maximize the value estimate on $s''$ while keeping close to the state distribution in the dataset. The prediction model $M_\xi(s)$ parameterized by $\xi$ is thus trained by minimizing:

$$\mathcal{L}_\xi = \mathbb{E}_{s,s' \sim \mathcal{D}}[\exp{(\beta A(s, s'))}\|s' - F_\phi(s, a')\|_2^2 - \alpha V_\psi(F_\phi(s, a'))], \tag{8}$$

where $a' = I_\omega(s, \hat{s}'), \hat{s}' = M_\xi(s)$. We follow [15] and set $\alpha = \frac{N}{\sum_{(s_i, s_i')} |Q(s_i, s_i')|}$ across all of the tasks we evaluate. Note that all of the models we describe above are deterministic.

## 4 Experiments

In this section, we examine the performance of our proposed SAW algorithm by conducting experiments on D4RL [14] datasets. We compare our SAW algorithm against some common baselines in offline RL, CQL [27], behavior cloning (BC), Decision Transformer (DT) [8], UWAC [55], TD3+BC [15], and IQL [24]. The results of CQL, UWAC and IQL are obtained by running their official codebase. We take results of TD3+BC and DT directly from its original paper. The results of BC are obtained by using our implemented code. All algorithms are run over 5 different random seeds. We present the comparison in Table 1, where we observe that SAW shows competitive or even better performance against the baseline methods.

For most of the MuJoCo tasks, we use temperature $\beta = 5.0$ and the expectile $\tau = 0.7$. We defer the offline-to-online experiments to Appendix D due to space limit where we find that SAW exhibits good generalization capability when transferring from offline to online. We also observe that SAW can learn some meaningful policies on challenging antmaze tasks, which can be found in Appendix D. The antmaze environment is very stochastic, and it is a typical sparse reward task. It is known that D3G requires a deterministic transition dynamics, while the success of our SAW on antmaze breaks the stereotype that QSS learning methods cannot succeed on stochastic environments.

Table 1: Normalized average score comparison of SAW against baselines on D4RL benchmarks over the final 10 evaluations. 0 corresponds to a random policy and 100 corresponds to an expert policy. The experiments are run on MuJoCo "-v2" datasets over 5 random seeds. r = random, m = medium, m-r = medium-replay, m-e = medium-expert, e = expert. We **bold** the first and second highest mean.

| Task Name | BC | DT [8] | CQL [27] | UWAC [55] | TD3+BC [15] | IQL [24] | SAW (ours) |
|---|---|---|---|---|---|---|---|
| halfcheetah-r | 2.2±0.0 | - | **17.5**±1.5 | 2.3±0.0 | 11.0±1.1 | 13.1±1.3 | **23.0**±3.9 |
| hopper-r | 3.7±0.6 | - | 7.9±0.4 | 2.7±0.3 | **8.5**±0.6 | **7.9**±0.2 | 7.3±0.6 |
| walker2d-r | 1.3±0.1 | - | 5.1±1.3 | 2.0±0.4 | 1.6±1.7 | **5.4**±1.2 | **5.6**±1.5 |
| halfcheetah-m | 43.2±0.6 | 42.6±0.1 | 47.0±0.5 | 42.2±0.4 | **48.3**±0.3 | 47.4±0.2 | **47.5**±0.3 |
| hopper-m | 54.1±3.8 | 67.6±1.0 | 53.0±28.5 | 50.9±4.4 | 59.3±4.2 | **66.2**±5.7 | **95.4**±5.1 |
| walker2d-m | 70.9±11.0 | 74.0±1.4 | 73.3±17.7 | 75.4±3.0 | **83.7**±2.1 | **78.3**±8.7 | 74.8±6.9 |
| halfcheetah-m-r | 37.6±2.1 | 36.6±0.8 | **45.5**±0.7 | 35.9±3.7 | **44.6**±0.5 | 44.2±1.2 | 43.9±0.5 |
| hopper-m-r | 16.6±4.8 | 82.7±7.0 | 88.7±12.9 | 25.3±1.7 | 60.9±18.8 | **94.7**±8.6 | **97.3**±2.8 |
| walker2d-m-r | 20.3±9.8 | 66.6±3.0 | **81.8**±2.7 | 23.6±6.9 | **81.8**±5.5 | 73.8±7.1 | 58.0±6.4 |
| halfcheetah-m-e | 44.0±1.6 | 86.8±1.3 | 75.6±25.7 | 42.7±0.3 | **90.7**±4.3 | 86.7±5.3 | **89.9**±4.9 |
| hopper-m-e | 53.9±4.7 | **107.6**±1.8 | **105.6**±12.9 | 44.9±8.1 | 98.0±9.4 | 91.5±14.3 | 90.0±7.7 |
| walker2d-m-e | 90.1±13.2 | 108.1±0.2 | 107.9±1.6 | 96.5±9.1 | **110.1**±0.5 | **109.6**±1.0 | 107.2±1.9 |
| halfcheetah-e | 91.8±1.5 | - | **96.3**±1.3 | 92.9±0.6 | **96.7**±1.1 | 95.0±0.5 | 95.4±0.8 |
| hopper-e | 107.7±0.7 | - | 96.5±28.0 | **110.5**±0.5 | 107.8±7 | **109.4**±0.5 | 102.6±6.6 |
| walker2d-e | 106.7±0.2 | - | 108.5±0.5 | 108.4±0.4 | **110.2**±0.3 | **109.9**±1.2 | 103.5±6.7 |

## 5 Conclusion

In this paper, we explore the potential of QSS learning in offline RL. We expect the agent can reach high reward states and the action is executed to ensure that the agent can reach the desired state. We

leverage *state advantage weighting* for guiding the actor and the prediction model. Our method, **S**tate **A**dvantage **W**eighting (SAW) shows good performance on D4RL datasets. We also demonstrate good generalization ability of SAW with additional online interactions after offline training. To the best of our knowledge, we are the first that makes QSS learning work in offline RL. We are very excited on this direction and are expecting further improvement of our algorithm in future work.

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

## A  Related Work

**Offline RL.** Offline reinforcement learning (or batch RL) [28, 30] aims at learning a well-behaved policy using only the fixed dataset that was previously collected by some unknown behavior policy. A unique challenge in offline RL lies in the bootstrapping error [16, 26], where the agent overestimates and prefers the OOD actions, often resulting in poor performance. Current solution class involves constraining the learned policy to be close to behavior policy [15, 26, 54, 51, 13, 56], learning completely within the dataset's support [24, 57, 19, 16, 61, 50, 9], adopting model-based methods [59, 22, 58, 38, 11], leveraging uncertainty measurement [2, 55, 38, 1], importance sampling [41, 45, 31, 36, 18], injecting conservatism into value learning [33, 27, 32, 25], sequential modeling [21, 8], etc.

**QSA and QSS Learning.** Traditionally, the Q-learning [44, 53] is conducted in the QSA style, i.e., $Q(s, a) \leftarrow Q(s, a) + \alpha \left[ r + \gamma \max_{a' \in \mathcal{A}} Q(s', a') - Q(s, a) \right]$. Many algorithms are constructed based on the QSA learning, which achieve remarkable success in both discrete control [35, 3, 52] and continuous control [20, 17]. Whereas, the QSS learning style is rarely studied. D3G [12] is probably the first QSS learning algorithm. D3G assumes that transition dynamics are deterministic, and shows that the QSS learning paradigm is equivalent to that of QSA learning under this assumption. Later on, there are attempts on learning from demonstration [6], improving online RL algorithms [60], planning [10], etc. We, however, aim at making QSS succeed in offline RL. We believe our work is of great value as it may provide good insights to learning from demonstration.

**Learning from Demonstration.** Learning from observation generally refers to learning meaningful policies without access to actions [42, 29, 49, 43, 47, 48]. It often aims at matching the performance of the expert policy [7]. Unlike this setting, we adopt QSS in offline RL where the agent may encounter the non-expert datasets, rising challenges for the algorithm to learn from. Meanwhile, our SAW does not require to match the trajectories in the dataset. Instead, it selects the good next state and encourages the agent to step towards that desired state.

## B  Missing Background on D3G

D3G [12] is a typical QSS learning algorithm. Note that all of the notations here are different from that in the main text. D3G has three components: the prediction model $\tau_\psi(s)$, the inverse dynamics model $I_\omega(s, s')$, and

the forward dynamics model $f_\phi(s, a)$ that are parameterized by $\psi, \phi, \omega$ respectively. D3G aims at learning in a QSS style while finding the next next state $s''$ in a neighboring state set $N(s)$, i.e.,

$$Q(s, s') \leftarrow Q(s, s') + \alpha[r + \gamma \max_{s'' \in N(s')} Q(s', s'') - Q(s, s')]. \tag{9}$$

$Q(s, s')$ is undefined if $s$ and $s'$ are not neighbors. $\tau_\psi(s)$ is a function that is used to select a good neighboring state that maximizes the state $Q$-function:

$$\tau_\psi(s) = \arg\max_{s' \in N(s)} Q(s, s'). \tag{10}$$

The action is then executed to make sure that the agent can reach the proposed state, $a = I_\omega(s, \tau_\psi(s))$. D3G only estimates the $Q$-function parameterized by $\theta$. The loss function for the critic is thus given by:

$$\mathcal{L}_{\theta_i} = \mathbb{E}_{s,s'} \|Q_{\theta_i}(s, s') - r - \gamma \min_{i=1,2} Q_{\theta_i'}(s', \tau_{\psi'}(s'))\|_2^2, \tag{11}$$

where $\theta_i'$ and $\psi'$ are the target parameter, $i = 1, 2$. With the predicted next state, we update the actor (or the reverse dynamics model) via imitation learning:

$$\mathcal{L}_\omega = \mathbb{E}_{s,a,s' \sim \mathcal{D}} \|I_\omega(s, s') - a\|_2^2. \tag{12}$$

To make sure that $\tau_\psi(s)$ always proposes the neighboring state that can be reached in one step. D3G regularizes $\tau_\psi(s)$ by additionally training a forward model. The forward model is trained via:

$$\mathcal{L}_\phi = \mathbb{E}_{s,a,s' \sim \mathcal{D}} \|f_\phi(s, a) - s'\|_2^2. \tag{13}$$

D3G leverages the inverse dynamics model and the forward dynamics model to ensure the consistency of the proposed state. To be specific, given a state $s$, D3G adopts the prediction model to propose a high reward state $\hat{s}'$, and then uses the reverse dynamics model $I_\omega(s, \hat{s}')$ to determine the action that would yield that transition. Then, the action is plugged into the forward model to get the next state $s_f'$. D3G then regularizes the deviation between $s_f'$ and $\hat{s}'$. The objective function for the prediction model is then given by:

$$\mathcal{L}_\psi = -\mathbb{E}_{s \sim \mathcal{D}}[Q_\theta(s, s_f')] + \mathbb{E}_{s \sim \mathcal{D}} \|\hat{s}' - s_f'\|_2^2. \tag{14}$$

D3G relies on a strong assumption that all of the transition dynamics are deterministic and they show that QSS learning is equivalent to QSA learning under such an assumption.

## C Experimental Setup and SAW Algorithm

In this section, we provide a detailed experimental setup for our proposed SAW algorithm and the baseline methods. We also include a detailed pseudo-code for the SAW algorithm.

### C.1 Experimental Details

#### C.1.1 D4RL datasets

We conduct our experiment mainly on D4RL [14] datasets, which are specially designed for the evaluation of offline RL algorithms. The D4RL datasets cover various dimensions that offline RL may encounter in practical applications in real world, such as passively logged data, human demonstrations, etc. The MuJoCo dataset in D4RL is collected during the interactions of different levels of policies with the continuous action environments in Gym [5] simulated by MuJoCo [46]. In the main text, we evaluate our method against baselines on three tasks in this dataset, *halfcheetah, hopper, walker2d*. Each task in the MuJoCo dataset contains five types of datasets, *random, medium, medium-replay, medium-expert, expert*. **random:** a large amount of data that is collected by a random policy. **medium:** experiences collected from an early-stopped SAC policy for 1M steps. **medium-replay:** replay buffer of a policy trained up to the performance of the medium agent. **expert:** a large amount of data gathered by the SAC policy that is trained to completion. **medium-expert:** a large amount of data by mixing the medium data and expert data at a 50-50 ratio.

We adopt the normalized average score metric that is suggested in D4RL for performance evaluation of offline RL algorithms. Suppose the expected return of the random policy is $J_r$ (reference min score), and the expected return of an expert policy is $J_e$ (reference max score), the expected return of the offline RL algorithm is $J_\pi$ after training on the given dataset. Then the normalized score $\tilde{C}$ is given by Equation (15).

$$\tilde{C} = \frac{J_\pi - J_r}{J_e - J_r} \times 100. \tag{15}$$

The normalized score ranges roughly from 0 to 100, where 0 corresponds to the performance of a random policy and 100 corresponds to the performance of an expert policy. We give the detailed reference min score $J_r$ and reference max score $J_e$ for MuJoCo datasets in Table 2, where all of the tasks share the same reference min score and reference max score across different types of datasets (e.g., random, medium, etc.).

Table 2: The referenced min score and max score for the MuJoCo dataset in D4RL.

| Domain | Task Name | Reference min score $J_r$ | Reference max score $J_e$ |
|--------|-----------|---------------------------|---------------------------|
| MuJoCo | halfcheetah | $-280.18$ | 12135.0 |
| MuJoCo | hopper | $-20.27$ | 3234.3 |
| MuJoCo | Walker2d | 1.63 | 4592.3 |

### C.1.2 Implementation and hyperparemeters

There are generally five components in the SAW algorithm: a value function $V_\psi(s)$ parameterized by $\psi$, a forward dynamics model $F_\phi(s,a)$ parameterized by $\phi$, double critics $Q_{\theta_i}(s,s')$ parameterized by $\theta_i, i = 1, 2$, a prediction model $M_\xi(s)$ parameterized by $\xi$, and an reverse dynamics model $I_\omega(s,s')$ parameterized by $\omega$. All of them are represented by deterministic neural networks, i.e., three-layer MLP networks. The hidden neural size is set to be 256 for all of them, and the activation function is `relu`. The learning rate for all of the learnable models is set to be $3 \times 10^{-4}$. We adopt a discount factor $\gamma = 0.99$. We list in Table 3 the detailed hyperparameters (i.e., temperature $\beta$ and expectile $\tau$) we adopt for SAW on MuJoCo tasks. We set $\beta = 5.0$ and $\tau = 0.7$ for most of the tasks. We find that our method is not sensitive to the temperature $\beta$, while the expectile $\tau$ can have comparatively larger impact on the performance of the agent. Please refer to more evidence in Appendix D.3.

The results of the baseline methods are obtained by running their official codebase, e.g., CQL (https://github.com/aviralkumar2907/CQL), UWAC (https://github.com/apple/ml-uwac), IQL (https://github.com/ikostrikov/implicit_q_learning), etc. All methods are run over 5 random seeds with their average normalized scores reported.

Table 3: The detailed hyperparameters setup for SAW on MuJoCo tasks. Normalization $\alpha = $ ✗ denotes that the value function is not normalized, and vice versa.

| Taks Name | temperature $\beta$ | expectile $\tau$ | normalization $\alpha$ |
|-----------|---------------------|------------------|------------------------|
| halfcheetah-random-v2 | 5.0 | 0.7 | ✗ |
| hopper-random-v2 | 5.0 | 0.7 | ✔ |
| walker2d-random-v2 | 5.0 | 0.7 | ✔ |
| halfcheetah-medium-v2 | 5.0 | 0.7 | ✔ |
| hopper-medium-v2 | 5.0 | 0.7 | ✔ |
| walker2d-medium-v2 | 5.0 | 0.7 | ✔ |
| halfcheetah-medium-replay-v2 | 5.0 | 0.7 | ✔ |
| hopper-medium-replay-v2 | 5.0 | 0.7 | ✔ |
| walker2d-medium-replay-v2 | 5.0 | 0.7 | ✔ |
| halfcheetah-medium-expert-v2 | 5.0 | 0.7 | ✔ |
| hopper-medium-expert-v2 | 5.0 | 0.3 | ✔ |
| walker2d-medium-expert-v2 | 5.0 | 0.7 | ✔ |
| halfcheetah-expert-v2 | 5.0 | 0.7 | ✔ |
| hopper-expert-v2 | 5.0 | 0.3 | ✔ |
| walker2d-expert-v2 | 5.0 | 0.7 | ✔ |

### C.2 Overall Algorithm

We present the overall algorithm of SAW in Algorithm 1.

## D Extensive Experimental Results on D4RL Datasets

In this section, we provide extensive evidence on the benefits of the proposed SAW algorithm by conducting experiments on more datasets. We examine the performance of the SAW on the highly stochastic AntMaze tasks. We observe that SAW can achieve some meaningful policies compared against prior methods. The success of SAW on AntMaze tasks also breaks the stereotype that the QSS learning algorithm fails on stochastic tasks (i.e., D3G makes a strong assumption that the transition dynamics must be deterministic). Furthermore, we provide detailed parameter study of SAW on some selected datasets and tasks.

**Algorithm 1** State Advantage Weighting (SAW)
___
1: Initialize value network $V_\psi$, critic networks $Q_{\theta_1}, Q_{\theta_2}$, forward dynamics model $F_\phi$, prediction
   model $M_\xi$ and actor network $I_\omega$ with random parameters
2: Initialize target networks $\theta_1' \leftarrow \theta_1, \theta_2' \leftarrow \theta_2$ and offline replay buffer $\mathcal{D}$.
3: **for** $t = 1$ to $T$ **do**
4:  Sample a mini-batch $B = \{s, a, r, s', d\}$ from $\mathcal{D}$, where $d$ is the done flag
5:  Update value function by minimizing Equation (3)
6:  Update critics by minimizing Equation (5)
7:  Update the reverse dynamics model (actor) by minimizing Equation (7)
8:  Update the forward dynamics model by minimizing Equation (6)
9:  Update the prediction model by minimizing Equation (8)
10:  Update target networks: $\theta_i' \rightarrow \tau\theta_i + (1 - \tau)\theta_i'$, $i = 1, 2$
11: **end for**
___

## D.1 Results on Stochastic Antmaze Datasets

AntMaze is a very challenging domain where the ant is required to reach the goal in a maze. This task is very stochastic with its reward signal very sparse, therefore making it super difficult to learn in offline. We would like to examine whether our SAW, a QSS learning algorithm, can learn some meaningful policies. It is challenging for QSS learning algorithms like SAW to fulfill that. The reasons lie in two aspects: (1) all of the networks in SAW are deterministic; (2) it is hard to predict a good next next state in a sparse reward and stochastic environment. We conduct experiments on 6 AntMaze datasets, and compare SAW against behavior cloning (BC), Decision Transformer (DT) [8], AWAC [37], Onestep RL [4], TD3+BC [15], CQL [27], and IQL [24]. We take the results of these baseline methods from [24]. The standard deviation is omitted since it is not reported in [24]. We run SAW on these datasets over 5 different random seeds, and report the average normalized score in conjunction with one standard deviation in Table 4.

We observe that SAW does learn meaningful policies on these stochastic tasks, and is competitive to prior methods on some datasets. Even in a larger map, e.g., antmaze-large-diverse, SAW can still learn some (probably) useful policies and knowledge. We thus conclude that QSS learning can also work in stochastic environments. However, we do find that the overall performance of SAW is worse than IQL or CQL, while we believe this is a good starting point and it is interesting to further improve the performance of the SAW. We list in Table 5 the hyperparameters we adopt on these tasks, where we unify parameters across all datasets.

Table 4: Normalized average score comparison of SAW against baselines on D4RL benchmarks over the final 10 evaluations. 0 corresponds to a random policy and 100 corresponds to an expert policy. The experiments are run on antmaze "-v0" datasets over 5 random seeds. We **bold** the first and second highest mean.

| Task Name | BC | DT | AWAC [37] | Onestep RL [4] | TD3+BC | CQL | IQL | SAW (ours) |
|---|---|---|---|---|---|---|---|---|
| antmaze-umaze-v0 | 54.6 | 59.2 | 56.7 | 64.3 | 78.6 | 74.0 | **87.5** | **86.7**±12.5 |
| antmaze-umaze-diverse-v0 | 45.6 | 53.0 | 49.3 | 60.7 | **71.4** | **84.0** | 62.2 | 52.5±25.9 |
| antmaze-medium-play-v0 | 0.0 | 0.0 | 0.0 | 0.3 | 10.6 | **61.2** | **71.2** | 12.5±13.0 |
| antmaze-medium-diverse-v0 | 0.0 | 0.0 | 0.7 | 0.0 | 3.0 | **53.7** | **70.0** | 22.5±22.8 |
| antmaze-large=play-v0 | 0.0 | 0.0 | 0.0 | 0.0 | 0.2 | **15.8** | **39.6** | 2.5±4.3 |
| antmaze-large-diverse-v0 | 0.0 | 0.0 | 1.0 | 0.0 | 0.0 | **14.9** | **47.5** | 10.0±0.0 |

Table 5: The detailed hyperparameters setup for SAW on AntMaze tasks.

| Taks Name | temperature $\beta$ | expectile $\tau$ | normalization $\alpha$ |
|---|---|---|---|
| antmaze-umaze-v0 | 50.0 | 0.9 | ✔ |
| antmaze-umaze-diverse-v0 | 50.0 | 0.9 | ✔ |
| antmaze-medium-play-v0 | 50.0 | 0.9 | ✔ |
| antmaze-medium-diverse-v0 | 50.0 | 0.9 | ✔ |
| antmaze-large=play-v0 | 50.0 | 0.9 | ✔ |
| antmaze-large-diverse-v0 | 50.0 | 0.9 | ✔ |

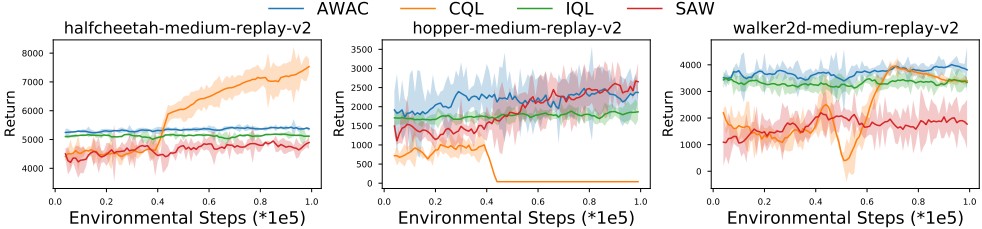

Figure 2: Offline-to-Online experiments on halfcheetah-medium-replay-v2, hopper-medium-replay-v2 and walker2d-medium-replay-v2. The results are averaged over 5 different random seeds.

## D.2    Offline-to-Online Experiments

We now investigate the offline-to-online adaption capability of the proposed SAW algorithm. We first conduct offline-to-online generalization test on three MuJoCo datasets, halfcheetah-medium-replay-v2, hopper-medium-replay-v2 and walker2d-medium-replay-v2. We compare SAW against AWAC [37], which is specially designed for offline-to-online adaption, CQL [27], and IQL [24]. For offline-to-online adaption, we gradually increase the ratio of newly interacted transitions with the iteration. That is, when we sample a batch of size $M$ samples for training, we sample $\eta M$ from offline replay buffer, and $(1 - \eta)M$ samples from the online buffer to mitigate the potential negative impact of distribution shift when transferring from offline to online. We set $\eta = 1 - \frac{t}{2T}$, where $t$ denotes the current iteration step and $T$ the total online iteration steps. That means, we increase the ratio of new samples with the iteration. We adopt $10^5$ steps of online interactions. We summarize the results in Figure 2. We find that SAW exhibits good performance on hopper-medium-replay-v2, and can hold the performance on halfcheetah-medium-replay-v2 and walker2d-medium-replay-v2.

We further conduct offline-to-online adaption on AntMaze domain, where we compare our SAW against AWAC, CQL and IQL. We run SAW over 5 different random seeds with additional 1M online interaction steps. We take the offline-to-online transfer results of AWAC, CQL and IQL from [24] directly. We summarize the overall results in Table 6. We find that on some of the datasets, SAW is able to keep the good offline performance, while it collapses on some datasets, e.g., antmaze-large-play-v0.

We note that it is very difficult for QSS learning methods to exhibit good performance when transferring from offline to online because of (1) there exists distribution shift between offline dataset and samples acquired during online interactions; (2) it is hard for QSS learning methods to propose a good next state that is reachable when interacting with the environment. If the proposed next state is poor or unreachable, the agent would fail to execute reasonable actions, resulting in poor performance. Whereas, we find that SAW can actually keep the performance with additional online interactions, or even learn better (e.g., hopper-medium-replay-v2). On hard tasks like AntMaze, SAW fails on some of them, which can be left as an interesting future direction to address. It is also very interesting to investigate how to output a reasonable high reward next state that can be reached in one step.

Table 6: Offline-to-online fine-tuning results on D4RL AntMaze datasets.

| Task Name | AWAC | CQL | IQL | SAW (ours) |
|---|---|---|---|---|
| antmaze-umaze-v0 | $56.7 \rightarrow 59.0$ | $70.1 \rightarrow 99.4$ | $86.7 \rightarrow 96.0$ | $77.5 \rightarrow 75.0$ |
| antmaze-umaze-diverse-v0 | $49.3 \rightarrow 49.0$ | $31.1 \rightarrow 99.4$ | $75.0 \rightarrow 84.0$ | $10.0 \rightarrow 36.7$ |
| antmaze-medium-play-v0 | $0.0 \rightarrow 0.0$ | $23.0 \rightarrow 0.0$ | $72.0 \rightarrow 95.0$ | $32.5 \rightarrow 10.0$ |
| antmaze-medium-diverse-v0 | $0.7 \rightarrow 0.3$ | $23.0 \rightarrow 32.3$ | $68.3 \rightarrow 92.0$ | $6.7 \rightarrow 3.3$ |
| antmaze-large=play-v0 | $0.0 \rightarrow 0.0$ | $1.0 \rightarrow 0.0$ | $25.5 \rightarrow 46.0$ | $0.0 \rightarrow 0.0$ |
| antmaze-large-diverse-v0 | $1.0 \rightarrow 0.0$ | $1.0 \rightarrow 0.0$ | $42.6 \rightarrow 60.7$ | $10.0 \rightarrow 15.0$ |

## D.3    Parameter Study

There are mainly two additional hyperparameters that are introduced in our SAW algorithm, the temperature $\beta$ and the expectile $\tau$. We examine the influence of these parameters by conducting experiments on two typical datasets from MuJoCo datasets, hopper-medium-replay-v2 and halfcheetah-medium-v2. We search $\beta$ among $\{0.5, 3.0, 5.0, 10.0\}$ and the expectile $\tau$ among $\{0.5, 0.7, 0.9\}$. We present the results below in Figure 3, where we find that our method is insensitive to the temperature $\beta$ (see Figure 3(a) and 3(b)), while the expectile $\tau$ can have comparatively larger impact (see Figure 3(c) and 3(d)). Smaller expectile tends to incur poor performance, while it seems that there always exists an intermediate value that can achieve the best trade-off. It is interesting

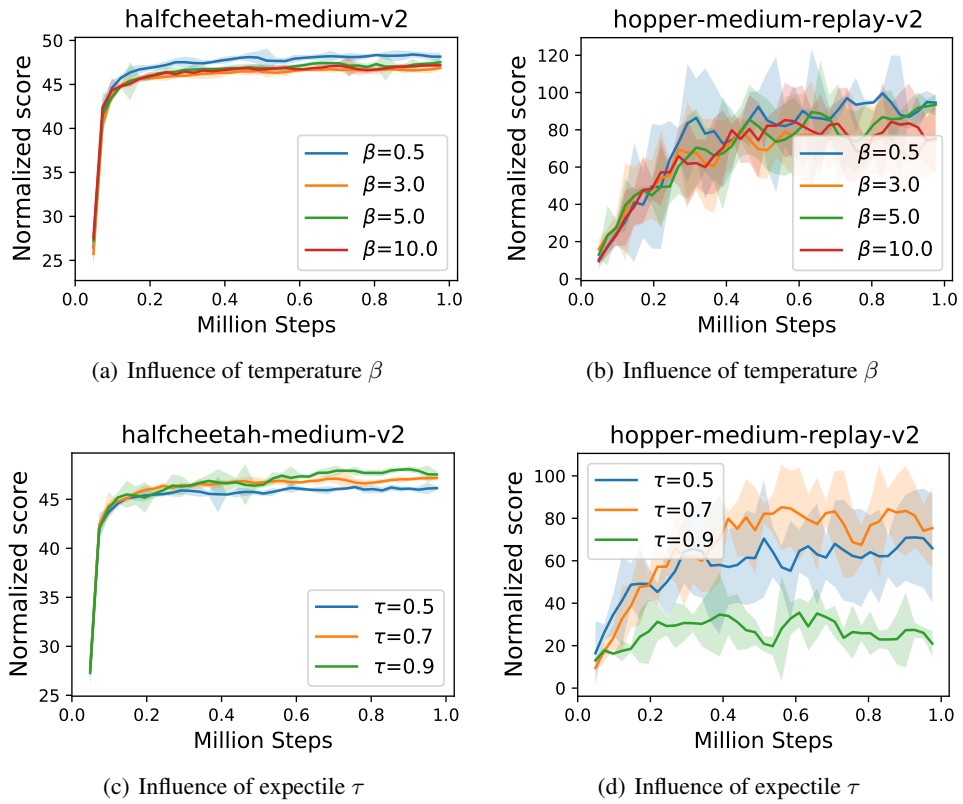

Figure 3: Parameter study of SAW on two selected datasets, halfcheetah-medium-v2 and hopper-medium-replay-v2. All experiments are run and averaged over 5 different random seeds and the shaded region denotes the standard deviation.

that the default we adopt for these datasets may not be optimal. For example, on halfcheetah-medium-v2, $\beta = 5.0, \tau = 0.9$ exhibit better performance than $\beta = 5.0, \tau = 0.7$; on hopper-medium-replay-v2, $\beta = 0.5$ may be better. We try to unify hyperparameters to show the advantages of our method and for simplicity.

