# OpenReview forum: "State Advantage Weighting for Offline RL"
_NeurIPS.cc/2022/Workshop/Offline_RL — Offline RL Workshop NeurIPS 2022_

### Official Review · Reviewer_xzqY · 2022-10-21
**This paper presents an algorithm for offline RL with state advantage learning.**

**Rating:** 5
**Confidence:** 3

**Review:**

The paper provides an interesting approach to solving the problem of overestimation of OOD actions, by decoupling them entirely and proposing a novel QSS-style learning paradigm for offline RL instead. The novelty of the QSS learning algorithm is to combine D3G with IQL-style expectile regression for the critic and (state) advantage-weighted regression for updating the prediction model and the actor. However, there is some explanation/motivation missing as to why D3G exhibits very poor performance on continuous control tasks with its vanilla formulation in regular RL settings, and why the offline RL formulism is helpful for D3G specifically. I am also unsure how this work differs from IQL other than using state advantage estimation. There is also some explanation missing as to why the new QSS-style method (SAW) performs better than the state-of-the-art method IQL.